# The microRNA Pathway of Macroalgae: Its Similarities and Differences to the Plant and Animal microRNA Pathways

**DOI:** 10.3390/genes16040442

**Published:** 2025-04-09

**Authors:** Jessica Webb, Min Zhao, Alexandra H. Campbell, Nicholas A. Paul, Scott F. Cummins, Andrew L. Eamens

**Affiliations:** 1Seaweed Research Group, University of the Sunshine Coast, Maroochydore, QLD 4558, Australiamzhao@usc.edu.au (M.Z.); acampbe1@usc.edu.au (A.H.C.); npaul@usc.edu.au (N.A.P.); scummins@usc.edu.au (S.F.C.); 2School of Science, Technology and Engineering, University of the Sunshine Coast, Maroochydore, QLD 4558, Australia; 3School of Health, University of the Sunshine Coast, Maroochydore, QLD 4558, Australia

**Keywords:** microRNA (miRNA), miRNA evolution, miRNA conservation, miRNA pathway, plant miRNA pathway, animal miRNA pathway, seaweed miRNA pathway

## Abstract

In plants and animals, the microRNA (miRNA) class of small regulatory RNA plays an essential role in controlling gene expression in all aspects of development, to respond to environmental stress, or to defend against pathogen attack. This well-established master regulatory role for miRNAs has led to each protein-mediated step of both the plant and animal miRNA pathways being thoroughly characterized. Furthermore, this degree of characterization has led to the development of a suite of miRNA-based technologies for gene expression manipulation for fundamental research or for use in industrial or medical applications. In direct contrast, molecular research on the miRNA pathway of macroalgae, specifically seaweeds (marine macroalgae), remains in its infancy. However, the molecular research conducted to date on the seaweed miRNA pathway has shown that it shares functional features specific to either the plant or animal miRNA pathway. In addition, of the small number of seaweed species where miRNA data is available, little sequence conservation of individual miRNAs exists. These preliminary findings show the pressing need for substantive research into the seaweed miRNA pathway to advance our current understanding of this essential gene expression regulatory process. Such research will also generate the knowledge required to develop novel miRNA-based technologies for use in seaweeds. In this review, we compare and contrast the seaweed miRNA pathway to those well-characterized pathways of plants and animals and outline the low degree of miRNA sequence conservation across the polyphyletic group known as the seaweeds.

## 1. The microRNA Pathway and Its Evolution in Eukaryotes

In eukaryotes, an expansive population of small regulatory RNAs (sRNAs) have been demonstrated to act as master controllers of gene expression functioning at either the transcriptional or posttranscriptional level. Moreover, sRNA-directed gene expression regulation has been shown to be essential for the normal progression of development, to respond to environmental stress, to defend against invading pathogens, and to maintain genome integrity via inhibiting the activity of mobile genetic elements [1,2,3,4,5,6,7]. The critical importance of sRNA-directed gene expression regulation as part of eukaryotic development is best evidenced when dysfunction of this regulatory mechanism occurs, namely: loss of control of sRNA-directed gene expression regulation has been associated with the onset of several disease states in numerous higher eukaryotes. For example, defective sRNA-directed gene expression regulation has been linked to the onset of specific types of cancer in humans [8] and for other mammalian species [9], chronic kidney disease in domesticated dogs [10], and mastitis in dairy cows [11]. Although an array of functionally distinct species of sRNA have been identified across evolutionarily diverse eukaryotes, sRNAs can typically be grouped into two primary classes, including the (1) microRNA (miRNA), and (2) small-interfering RNA (siRNA) classes [12,13,14]. Grouping sRNAs into these two classes is based on the structure of the double-stranded RNA (dsRNA) precursor transcript from which the sRNA is liberated during the production stage of its functional maturation into a sRNA silencing signal [12,15,16,17]. More specifically, a single or predominant miRNA silencing signal is liberated from a stem-loop structured precursor composed of paired stem arms (i.e., nucleotides in the 5′ arm base pair with nucleotides in the 3′ arm) and an unpaired loop region: a dsRNA folding structure that is the result of incomplete complementarity of the non-coding RNA (ncRNA) precursor [12,15]. In contrast, multiple siRNA silencing signals, sometimes numbering in the hundreds, are processed from molecules of perfectly dsRNA, with such precursors being naturally generated in eukaryote nuclear genomes by the transcription of unique DNA structures, such as the convergent bidirectional transcription of repetitive DNA elements [16,17].

Post liberation from the dsRNA precursor, and their subsequent functional maturation, miRNAs and siRNAs form single-stranded ncRNAs of a predominant length of 20 to 24 nucleotides (20–24-nt) [12,13,14,15,16,17]. This size, together with the structure and genomic origin of the precursor, determines the loading of each sRNA species into a specific protein effector complex, termed the RNA-induced silencing complex (RISC), which facilitates the regulatory mechanism of gene expression control directed by the loaded sRNA [18,19,20,21,22]. For example, in plants, miRNAs are primarily in the form of a 21-nt silencing signal which directs their preferential loading into a protein effector complex that mediates target gene expression regulation via a messenger RNA (mRNA) cleavage mode of RNA silencing [18,19,20]. In animal species however, miRNA silencing signals are primarily 22-nt in length upon their functional maturation, a size which guides their directed loading into a protein effector complex that controls the expression of target gene transcripts via a translational repression mode of RNA silencing [21,22]. Protein effector complex loading of miRNAs in both the plant and animal systems, activates the complex to form, miRNA-loaded RISC (miRISC) [19,22]. Plant miRNAs display a high degree of complementarity to the target gene transcripts whose expression is regulated by miRISC. Therefore, each plant miRNA only directs expression regulation of a small number of closely related target genes which encode proteins of highly similar molecular function [23,24]. In direct contrast, in the animal system, the miRNA silencing signal only has limited complementarity at its 5′ end, termed the ‘*seed sequence*’ to its target gene transcripts, thereby facilitating the ability of each miRISC-loaded miRNA to influence the expression of a considerably higher number of target genes, target genes that encode protein products of completely unrelated molecular function [25,26].

Since their initial discovery in *Caenorhabditis elegans* (roundworm) in 1993 [27], miRNAs have been identified and experimentally validated across the eukaryotes, from the unicellular green microalgae, *Chlamydomonas reinhardtii* (*Chlamydomonas*) [28] to the genetic model plant species *Arabidopsis thaliana* (*Arabidopsis*) [29], the widely used species for developmental biology studies *Drosophila melanogaster* (fruit fly) [30] and *Mus musculus* (mouse) [31], through to humans (*Homo sapiens*) [32]. Profiling of the miRNA landscape of a wide breadth of eukaryotes has shown a comprehensive lack of sequence homology between the predominant miRNA families encoded by plant and animal nuclear genomes. This, in addition to the mechanistic differences in the production and action stages of the plant and animal miRNA pathways, strongly suggests that miRNAs have evolved independently from an ancient sRNA pathway, most likely a siRNA-like pathway, that already existed in the last common ancestor of these two predominant eukaryote kingdoms [27,28,29,30,31].

The notion that the miRNA pathway has evolved largely independently in individual eukaryotic lineages is further supported by the only recent identification of shared miRNA species between members of the two main divisions of the Viridiplantae, algae and plants. Namely, three miRNAs of similar sequence identity were discovered in the green microalgae *C. reinhardtii*, and the thalloid liverwort species, *Pellia endiviifolia* [32,33]. This recent identification of miRNAs shared in algae and plants emphasizes the importance of sampling, at an adequate depth of coverage, the miRNA landscape of many species from each phylum to confidently identify the degree of miRNA conservation across the eukaryotes [32,33,34,35]. Sampling each species at an adequate depth of coverage, as well as sampling multiple developmental stages and different growth environment conditions of the species under assessment, not only allows for the identification of the full miRNA repertoire of a species, but aids in improving the possibility of uncovering the full degree of miRNA conservation in the eukaryotes [36,37,38]. Such an approach also helps to address the inherent difficulty in confidently annotating both conserved and species-specific miRNAs (also termed ‘*newly evolved miRNAs*’) in each species under analysis [39,40,41,42] with the adoption of this cautious approach to miRNA pathway analysis adding further weight to the prevailing view that the miRNA pathway has evolved convergently in plants and animals.

In the eukaryotes, the extensive miRNA pathway data available today strongly suggests that the miRNA pathway has likely evolved independently at least nine times in the excavates, including in slime molds, brown algae, green algae, land plants, twice in the sponges (e.g., demosponge—*Amphimedon queenslandica* and calcareous sponge *Sycon ciliatum*), cnidarians, and the bilaterians [28,33,35,42,43,44,45,46,47,48]. Alternatively, the lack of miRNA sequence conservation between plants and animals could be accounted for by high sequence turnover rates which would have removed any trace of miRNA conservation in these two contemporary lineages [35,42,44,45,48]. As example, only a single conserved miRNA has been identified between *C. reinhardtii* (unicellular) and *Volvox carteri* (multicellular), two species of green microalgae that diverged from each other approximately (~) 200 million years ago (Mya) [28,33,49]. Similarly, albeit more surprising, profiling of the miRNA landscapes of *A. thaliana* and *A. lyrata* revealed that only 66% of *MICRORNA* (*MIR*) gene families are conserved between these two closely related species of *Arabidopsis* which only diverged from each other ~10 Mya [13,29,36,41].

In spite of the lack of miRNA sequence conservation, many studies have shown that the genomes of species capable of producing miRNAs encode (1) Dicer (Dcr) or DICER-LIKE (DCL), the RNase III-like endonuclease required to process the miRNA from its dsRNA precursor, and (2) ARGONAUTE (AGO), an endonuclease which forms the catalytic core of miRISC to mediate miRNA-directed target gene expression regulation, post miRNA maturation [16,17,18,19,20,21,22]. Therefore, for eukaryote species with available nuclear genome sequence data, but where miRNA profiling has not yet been performed, the first step in miRNA biology research should be the bioinformatic identification of genomic loci that encode for the Dcr/DCL and AGO endonucleases. Such an approach has formed the basis for construction of a detailed molecular understanding of the miRNA pathways of plants, animals and microalgae via the use of experimental model species such as *Arabidopsis*, *Drosophila* and *Chlamydomonas*, respectively [16,17,18,19,50,51,52,53]. Furthermore, detailed functional characterization of the miRNA pathway in plants, animals and microalgae has enabled the development of novel gene expression manipulation technologies such as the artificial miRNA (amiRNA), molecular sponge, and target mimic approaches: molecular tools which can be used in either the research setting to further characterize miRNA function, or in industry to manipulate the biosynthesis pathways of consumer market desired natural products. In stark contrast to the plant, animal and microalgal systems, research on miRNA biology in macroalgae, specifically the seaweeds (marine macroalgae), included within the green (Chlorophyta), brown (Phaeophyceae), and red (Rhodophyta) divisions of algae, remains in its infancy. In this review, we describe the well characterized plant, animal and microalgae miRNA pathways using *Arabidopsis*, *Drosophila*, and *Chlamydomonas* as our respective model species. We then detail the scarce findings reported to date on either the global miRNA populations or the core protein machinery of the miRNA pathways of green, brown, and red seaweeds and compare these preliminary findings with those reported in plants, animals, and microalgae. Further, we go on to outline the requirement to develop miRNA-based technologies for further functional characterization of the seaweed miRNA pathway and for use as a biotechnological tool to manipulate the biosynthesis pathways of consumer-desired natural products specific to the seaweeds.

## 2. The miRNA Pathway of *Arabidopsis thaliana*

The majority of *Arabidopsis MIR* genes occupy intergenic regions of the *Arabidopsis* nuclear genome, and which share structural features identical to those of protein-coding genes, including housing both a promoter region and terminator sequence within the gene body [54,55]. Due to these shared structural features, RNA polymerase II (Pol II; the same RNA polymerase responsible for protein-coding gene transcription in *Arabidopsis*) transcribes a ncRNA from the *MIR* gene, however, the nascent transcript does not contain either a translation start or stop codon, and due to these missing signals, the ncRNA will not be used as a translation template [54,55]. Pol II catalyzed transcription of the *MIR* gene derived ncRNA does nonetheless identify the ncRNA for posttranscriptional modifications, including the addition of a 7-methylguanosine capping structure at the 5′ terminus (5′ cap) and a polyadenylated tail at the 3′ terminus (3′ poly(A) tail) of the transcript [56,57]. In addition, due to the long length of *Arabidopsis MIR* gene transcription products, some also contain intron sequences, which identifies these miRNA precursors for further posttranscriptional modification via intron splicing [58,59]. Each *MIR* gene transcription product contains a region of partial sequence self-complementarity which directs this region of the ncRNA to fold back onto itself to form a stem-loop structured imperfectly dsRNA termed the primary-miRNA (pri-miRNA) [54,55]. In the *Arabidopsis* cell nucleus, the zinc-finger protein with RNA-binding capacity, SERRATE1 (*Ath*-SE1), recognizes and binds the stem-loop structured pri-miRNA and transports the bound molecule to specialized nuclear bodies, termed D-bodies (Dicing-bodies), where the Dicing complex (D-complex; also called the microprocessor complex) proteins localize to [60,61] (Figure 1).

In the D-body, the *Ath*-SE1 bound pri-miRNA undergoes a two-step processing event catalyzed by the core D-complex proteins, DICER-LIKE1 (*Ath*-DCL1) and dsRNA BINDING1 (*Ath*-DRB1) [60,61,62,63]. Initially, the *Ath*-DCL1 endonuclease cleaves off the unpaired, single-stranded RNA regions (i.e., removal of ncRNA sequences which flank the dsRNA stem-loop structure) of the pri-miRNA to produce the smaller sized processing intermediate, the precursor-miRNA (pre-miRNA). Next, *Ath*-DCL1 cleaves the pre-miRNA at specific positions within the molecule to remove a section of the paired stem arms, and the unpaired loop region of the pre-miRNA to generate the much smaller sized dsRNA molecule, the miRNA/miRNA* duplex [64,65,66,67] (Figure 1). RNase IIIa and RNase IIIb, the Dicing domains of *Ath*-DCL1, are positioned 21-nt apart from each other on DCL1-bound dsRNA substrates, and accordingly, most *Arabidopsis* miRNA silencing signals are 21-nt in length post precursor transcript processing by the D-complex. *Arabidopsis* DCL1 possesses two dsRNA binding domains (dsRBDs) at its carboxyl terminus (C-terminus), and therefore, can bind and process miRNA precursors on its own. However, both the accuracy (i.e., the position of cleavage) and efficiency (i.e., the rate of cleavage) of *Ath*-DCL1-catalyzed pri-miRNA and pre-miRNA processing is enhanced via *Ath*-DCL1 functionally interacting with the other D-complex core proteins, *Ath*-SE1 and *Ath*-DRB1 [60,61,64,66,68]. Moreover, during the precursor processing phases of the production stage of the *Arabidopsis* miRNA pathway, the primary role conducted by *Ath*-DRB1 is to accurately position *Ath*-DCL1 on the precursor transcript to ensure that just a single miRNA/miRNA* duplex is liberated from the much longer length pri-miRNA and pre-miRNA precursors by *Ath*-DCL1 dicing [64,65,68]. *Ath*-DCL1 cleavage generates a 2-nt overhang at the 3′ terminus of both strands of the miRNA/miRNA* duplex, and this dicing signature is recognized by the sRNA-specific methyltransferase, HUA ENHANCER1 (*Ath*-HEN1) (Figure 1). *Ath*-HEN1 adds a methyl group to the 2′ hydroxyl group (2′-O-methylation) of the 3′ terminal nucleotide of both the miRNA guide strand and miRNA* passenger strand while the two duplex strands are still hybridized to each other [69,70]. *Ath*-HEN1 similarly modifies the 3′ terminal nucleotide of each duplex strand of each class of siRNA which also accumulates in *Arabidopsis* cells. This universal sRNA modification is thought to protect the 3′ end of sRNAs post duplex strand separation from uridylation, and therefore degradation, to thus ensure that the sRNA retains its ‘*functionality*’ prior to its loading into its specific effector complex (i.e., miRISC and siRISC) [69,70].

**Figure 1 genes-16-00442-f001:**
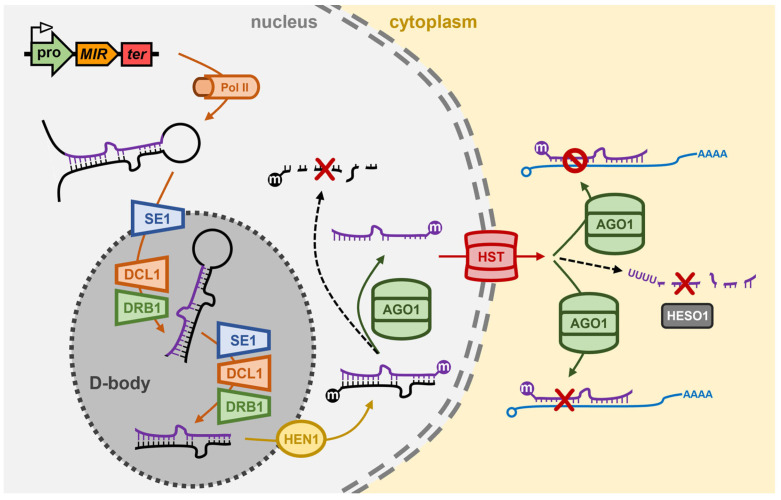
The miRNA pathway of the model plant *Arabidopsis thaliana*. In the nucleus of the *Arabidopsis* cell, Pol II transcribes the pri-miRNA from a *MIR* gene. The pri-miRNA is bound by SE1 and is transported to the D-body where it is processed by DCL1/DRB1 into a pre-miRNA and then a miRNA/miRNA* duplex. HEN1 methylates the 3′ terminus of both duplex strands and AGO1 retains the miRNA guide strand and discards the miRNA* passenger strand. HST is involved in the nucleus-to-cytoplasm export of some miRNAs, and in the cytoplasm of the *Arabidopsis* cell, AGO1 uses its loaded miRNA to guide the silencing of target gene transcripts predominantly via mRNA cleavage but also via translational repression. HESO1 controls the steady-state levels of *Arabidopsis* miRNAs to add an additional layer of complexity to miRNA-directed gene expression regulation. AGO1, ARGONAUTE1; D-body, nuclear Dicing body; DCL1, DICER-LIKE1; DRB1, dsRNA BINDING1; HEN1, HUA ENHANCER1; HESO1, HEN1 SUPPRESSOR1; *MIR*, *MICRORNA* gene; Pol II, RNA polymerase II; pro, *MIR* gene promoter; SE1, SERRATE1; ter, *MIR* gene terminator.

Post 2′-O-methylation of both duplex strands, *Ath*-DRB1 orientates the loading of the miRNA/miRNA* duplex into the ARGONAUTE1 (*Ath*-AGO1) endonuclease to ensure that *Ath*-AGO1 retains the miRNA guide strand and ‘*slices*’ the miRNA* sequence to remove the passenger strand from miRISC [18,19,71] (Figure 1). The now single-stranded and functionally mature miRNA, either on its own, or in complex with *Ath*-AGO1, is then exported out of the nucleus and into the cytoplasm of the *Arabidopsis* cell. HASTY (*Ath*-HST), the *Arabidopsis* ortholog of the animal RanGTP-dependent dsRNA-binding protein, Exportin-5 (XPO5), was originally thought to be required for the export of all *Arabidopsis* miRNAs from the nucleus to the cytoplasm [72,73] (Figure 1). However, more recent research has questioned the exact degree of requirement of *Ath*-HST for the nucleus export of mature miRNA silencing signals with the level of only some miRNAs reduced in the *Arabidopsis hst* mutant [72,73]. In the cytoplasm, *Ath*-AGO1 forms the catalytic core of miRISC for almost all *Arabidopsis* miRNAs (Figure 1), with each miRNA silencing signal loaded by *Ath*-AGO1 (and by the AGO1 homologs of other plant species) displaying a high degree of sequence complementarity to its target genes [18,19]. Furthermore, the target sites of miRISC-loaded miRNAs are predominantly located within the coding sequence of target genes, and together the (1) high degree of sequence complementarity between the miRNA and its targeted gene(s), and (2) coding sequence location of miRNA target sites, results in mRNA cleavage forming the primary mechanism of target gene expression regulation directed by an *Arabidopsis* miRNA [18,19]. Although much less prevalent than the target mRNA cleavage mechanism of miRNA-directed RNA silencing, more recent research has shown that many *Arabidopsis* miRNAs also mediate a translational repression mode of gene expression repression [74,75]: a process facilitated by protein machinery additional to the core machinery proteins described above, and which therefore sits outside of the scope of this review.

## 3. The miRNA Pathway of *Drosophila melanogaster*

Like in *Arabidopsis*, Pol II appears to be the primary DNA-dependent RNA polymerase responsible for *MIR* gene transcription in *Drosophila* [76,77,78]. However, in contrast to *Arabidopsis* where *MIR* genes are predominantly positioned between protein-coding genes, ~35% of *Drosophila MIR* genes are located proximal to one another on *Drosophila* chromosomes to form *MIR* gene clusters, and as such, are co-transcribed by Pol II to form polycistronic pri-miRNAs [79,80] (Figure 2). A further difference between the *MIR* gene genomic landscapes of *Arabidopsis* and *Drosophila* is that an additional ~30% of *Drosophila* miRNA precursor transcripts overlap with the sequences of protein-coding genes, with the precursors of this subclass of miRNAs, termed mirtrons, only forming post their release from the precursor-mRNA transcribed from the host gene following intron splicing [81,82] (Figure 2). Their positioning within protein-coding sequences results in mirtron expression being reliant on the expression of the host gene, and not on the unique composition of transcription regulating DNA-based elements of the *MIR* genes which occupy a position distinct to other gene loci on *Drosophila* chromosomes [78,83]. Post splicing, a mirtron containing intron folds to form a stem-loop structured dsRNA which is recognized for processing as a miRNA precursor transcript (i.e., recognized as a pri-miRNA) by the nucleus-localized RNase III-like endonuclease, *Dme*-Drosha [84,85] (Figure 2). As for *Ath*-DCL1, *Dme*-Drosha cleaves the single-stranded regions of the RNA transcript that flank the stem-loop folding structure of the pri-miRNA to form the pre-miRNA, which in *Drosophila* and other animal species, are dsRNA molecules of a uniform size of 60–70-nt and which harbor shared structural features of a stem-loop, an exposed phosphate group at the 5′ terminus, and a 2-nt overhang at the 3′ terminal end [84,85]. *Dme*-Drosha functionally interacts with the dsRNA binding protein, *Dme*-Pasha (also named DiGeorge syndrome critical region 8 (DGCR8)) to form the microprocessor complex in the nucleus of the *Drosophila* cell, with *Dme*-Pasha promoting both the accuracy and efficiency of *Dme*-Drosha catalyzed dicing of monocistronic, polycistronic and mirtronic pri-miRNAs [84,86,87] (Figure 2). During the nucleus-localized precursor processing stage of the *Drosophila* miRNA pathway, the RNA-binding protein Arsenic resistance 2 (*Dme*-Ars2) directs an analogous role to that of *Ath*-SE1, via *Dme*-Ars2 binding and stabilizing pri-miRNA precursors and promoting their interaction with the core protein machinery of the microprocessor complex, *Dme*-Drosha and *Dme*-Pasha, to ensure efficient pre-miRNA production [84,85,86,87,88] (Figure 2).

The pre-miRNA is next exported out of the nucleus and released into the cytoplasm by the action of the Ran-GTP-dependent dsRNA binding protein, *Dme*-XPO5 [89,90] (Figure 2). In the cytoplasm of the *Drosophila* cell, and once released by the *Dme*-Ran-GTP/*Dme*-XPO5 complex, the pre-miRNA is further processed by the cytoplasm-localized Dicer class of RNase III-like endonuclease, Dicer1 (*Dme*-Dcr1) [91,92,93]. Like *Dme*-Drosha, both the accuracy and efficiency of *Dme*-Dcr1 processing of pre-miRNA is enhanced via its interaction with its functional partner, the dsRNA binding protein, Loquacious (Loqs) (also called Transactivation response element RNA-binding protein (TRBP)) [92,94]. *Dme*-Dcr1/*Dme*-Loqs-directed processing of the pre-miRNA consists of removal of an unrequired section of the stem arms and the loop region of the stem-loop structure of the pre-miRNA, to generate a predominantly 22-nt miRNA/miRNA* duplex with 2-nt overhangs at the 3′ end of each duplex strand [91,92,93,94] (Figure 2). Typically, sequencing of the miRNA repertoire of an animal species has revealed that the abundance of the miRNA* strand is much lower than the level of the corresponding miRNA strand, usually at least ~100-fold lower, to suggest that preferential duplex strand selection is also a predominant biogenesis mechanism in animals, as it is in the plant model *Arabidopsis*. However, in contrast to *Arabidopsis* where *Ath*-DRB1 is responsible for orientating the miRNA/miRNA* duplex into *Ath*-AGO1 for miRNA guide strand selection, the *Dme*-Dcr1 functional partner protein of the cytoplasmic microprocessor complex, *Dme*-Loqs, appears not to function in an analogous manner at this stage in the *Drosophila* miRNA production pathway [71,92,94]. It is apparent however that the thermodynamic properties at the terminal base pair of the 5′ end of each duplex strand guides the orientation of the miRNA/miRNA* duplex into *Dme*-Ago1 for miRNA strand selection via unwinding of the duplex strands, retainment of the miRNA guide strand by *Dme*-Ago1, and removal of the miRNA* passenger strand for its subsequent degradation [95,96,97] (Figure 2). 

A member of the AGO protein family has also been shown to form the catalytic core of miRISC in other animal species, including *C. elegans*, mouse and humans, however, animal AGOs which are preferentially loaded with miRNA silencing signals are not cleavage competent, lacking the ‘*Slicer*’ activity of *Ath*-AGO1 [18,19,95,96]. Therefore, due to the lack of slicing ability by miRNA-loaded animal AGOs, translational repression forms the predominant mechanism of miRISC-mediated, miRNA-directed target gene expression regulation in animals [98,99,100]. Translational repression is a mild form of RNA silencing compared to the target transcript cleavage mode of RNA silencing directed by plant miRNAs. However, in the animal system, miRNA-guided target recognition is based on a low degree of miRNA/target mRNA complementarity with only miRNA nucleotide positions 2 to 8 required for an animal miRISC to use its loaded miRNA cargo as an identity guide to target mRNAs for expression regulation [101,102]. In addition to this, the target sites of animal miRNAs are almost exclusively positioned in the 3′ UTRs of miRISC-regulated mRNAs [103,104]. Therefore, these two parameters specific to the animal miRNA pathway results in (1) each miRNA identifying numerous unrelated transcripts for expression regulation, and (2) many of the targeted mRNAs being under co-regulation by two or more miRNAs, which would together, enhance or strengthen the overall degree of control on target gene expression regulation by the translational repression mode of RNA silencing. The PIWI domain of *Dme*-Ago1 mediates its interaction with the scaffolding protein GW182 via binding to the glycine/tryptophan (GW) repeats housed in the amino terminal (N-terminal) region of *Dme*-GW182 [105,106,107] (Figure 2). Interaction with *Dme*-GW182 provides a link between *Dme*-Ago1-mediated translational repression and de-adenylation complexes which in turn direct the removal of 3′ poly(A) tails and decay of miRISC bound mRNAs [105,106,107].

## 4. The miRNA Pathway of *Chlamydomonas reinhardtii*

The presence of miRNA silencing signals was first reported in *Drosophila* and *Arabidopsis* in 2001 [30] and 2002 [29], respectively, yet this class of regulatory RNA was not described in the unicellular green microalgae *C. reinhardtii* until 2007 following the publication of two studies [28,33], with some of the identified miRNAs reported by both studies. The [33] study identified 68 candidate miRNAs, of which 21 were further shown to originate from precursor molecules which could form stem-loop folding structures like those adopted by plant and animal miRNA precursors. Further confidence that a bona fide set of *Chlamydomonas* miRNAs had been identified was provided by the authors reporting that a miRNA* passenger strand in addition to a miRNA guide sequence was detected for 12 of the 21 miRNAs where a precursor molecule had also been identified [33]. The [28] study revealed that the predominant size of *Chlamydomonas* miRNAs was 21-nt like *Arabidopsis* miRNAs, however, genome mapping of the identified miRNAs showed that these sequences aligned to both the introns of protein-coding genes (~30%) and intergenic regions (~70%) of *Chlamydomonas* chromosomes, similar to the composition of the *MIR* gene landscape in *Drosophila* [28]. The authors also reported that the expression of some of the identified miRNAs was different between the vegetative and gametic cell forms of *Chlamydomonas*, and via interpretation of their in vitro analyses, this early report proposed that target transcript cleavage was the likely mode of RNA silencing directed by *Chlamydomonas* miRNAs [28]. Interestingly, although both reports [28,33] revealed that the *Chlamydomonas* miRNA pathway shared features specific to either the plant or animal miRNA pathway, no sequence similarity was found for the newly identified *Chlamydomonas* miRNAs to those previously identified for any plant or animal species whose miRNA landscape had been profiled and deposited into the miRNA Registry, miRBase (https://www.mirbase.org/) [108].

*Chlamydomonas reinhardtii* encodes three *DCL* genes, *Cre-DCL1* to *Cre-DCL3*, and considering that *Chlamydomonas* diverged from the last common ancestor of land plants over 1 billion years ago [109], it is unsurprising that the three *Chlamydomonas DCL*s are not direct orthologues of the *Ath-DCL1* to *Ath-DCL3* genes [110]. Via the use of amiRNA technology, and the mapping of the causative mutation of transformant lines defective in amiRNA-directed silencing of the endogenous gene *PHYTOENE SYNTHASE* (*PSY*), ref. [111] identified *Cre*-DCL3, a Dcr more structurally similar to the animal Drosha enzyme than to Dcr, as the DCL family member most likely responsible for miRNA production in *Chlamydomonas* [111]. More specifically, in the *Chlamydomonas dcl3* mutant, the amiRNA did not accumulate to detectable levels as it failed to be processed from the precursor which was developed to deliver the *PSY*-targeting amiRNA. Accordingly, amiRNA-specific cleavage products could not be detected in the *dcl3* mutant and *PSY* expression remained at its wild-type level [111]. In addition to the amiRNA failing to accumulate to detectable levels in the *Chlamydomonas dcl3* mutant, the clear 21-nt peak of the global sRNA population observed in wild-type cells by sRNA sequencing was absent in the mutant background with endogenous miRNA accumulation shown to be globally reduced in the *dcl3* mutant. Moreover, miRNA accumulation was reduced in the *Chlamydomonas dcl3* mutant regardless of the genomic origin of the endogenous miRNAs, with the levels of *MIR* gene-derived miRNAs, mirtron miRNAs, and miRNAs derived from the 3′ UTRs of host genes, all reduced in their abundance [111]. Further evidence that *Cre*-DCL3 is the DCL family member responsible for miRNA production in *Chlamydomonas* was provided by the authors’ demonstration that the *dcl3* mutation could be complemented via the introduction of a *Chlamydomonas* chromosome fragment which encoded a wild-type copy of the *Cre-DCL3* gene. In the complemented transformant lines, endogenous miRNAs accumulated to their wild-type levels, miRNA-specific target transcript cleavage products were readily detectable, and the expression of known miRNA target genes returned to wild-type equivalent levels [111]

A second transgene-based screen for mutants defective in miRNA-directed RNA silencing of a reporter gene, identified the *dull slicer-16* (*dus16*) mutant [112]. Mapping of the causative mutation of the *dus16* mutant revealed that the disrupted gene, *Cre05.g232004*, encoded an RNA binding protein which harbors both an RNA-binding motif (RBM) and a dsRBD: a functional domain configuration which would enable the encoded DUS16 protein to recognize and bind both single-stranded and double-stranded forms of RNA [112]. Moreover, and as shown for the *Chlamydomonas dcl3* mutant [111], pri-miRNA transcripts generated by the activity of Pol II over-accumulated, and 21-nt miRNA silencing signals were reduced to almost undetectable levels, in the *dus16* mutant background [112]. The construction of reporter tagged versions of *Cre*-DUS16 revealed that *Cre*-DUS16 predominantly localized to the nucleus with a smaller abundance of the tagged protein version localizing to cytoplasmic bodies. In addition, mass spectrometry analysis revealed that *Cre*-DUS16 was associated with *Cre*-DCL3, the predominant DCL family member responsible for miRNA production in *Chlamydomonas* [112]. Further characterization of *Cre*-DUS16 showed that it not only interacts with *Cre*-DCL3, but that *Cre*-DUS16 is required by *Cre*-DCL3 to enhance both the accuracy and efficiency of *Cre*-DCL3-catalyzed processing of miRNA precursor transcripts in *Chlamydomonas* [113] (Figure 3). This finding provided the first definitive evidence that *Cre*-DCL3/*Cre*-DUS16 form the microprocessor complex in the nucleus of *Chlamydomonas* cells and which functions in an analogous manner to the nucleus-localized microprocessor complexes DCL1/DRB1 in *Arabidopsis* [64,65], and Drosha/Pasha in *Drosophila* [84,85,86,87]. It is important to note here that, to date, no *Ath*-SE1 homolog has been described for *Chlamydomonas*, however, the identification of a RBM and a dsRBD in the *Cre*-DUS16 protein raises the intriguing proposal that *Cre*-DUS16 may function as both *Ath*-SE1 and *Ath*-DRB1 of the *Arabidopsis* microprocessor complex in the production stage of the *Chlamydomonas* miRNA pathway [112].

In addition to encoding three DCLs, *Chlamydomonas* encodes three AGO proteins, *Cre*-AGO1 to *Cre*-AGO3 [110], and in the *dcl3* and *dus16* mutants where miRNA production is largely defective [111,112], the abundance of the cytoplasmic AGO, *Cre*-AGO3, was shown to be almost undetectable. This finding indicated that in the absence of loading of its preferred cargo, miRNA silencing signals, the *Cre*-AGO3 protein was destabilized, and therefore likely to be rapidly turned over in the cell [114]. Similarly, in the *ago3-1* mutant, global miRNA abundance is reduced [114] to (1) indicate that mature miRNA silencing signals are rapidly turned over in the *Chlamydomonas* cell in the absence of loading into *Cre*-AGO3, and (2) further identify *Cre*-AGO3 as the primary AGO protein family member required to mediate miRNA-directed RNA silencing in *Chlamydomonas*. Protein pull-down experimentation and subsequent sequencing of the low molecular weight fraction to profile the sRNA landscape of *Cre*-AGO3, further showed that the majority of loaded sRNAs were between 20- to 22-nt in length, with most of these being 21-nt in length and which harbored a 5′ terminal uracil (U) residue [114]. This analysis indicated that like *Ath*-AGO1 [18,19], *Cre*-AGO3 has a strong loading preference based on sRNA size and starting nucleotide composition [114]. 

The initial characterization of miRNA-directed RNA silencing in *Chlamydomonas*, including the use of amiRNAs designed to have an extremely high degree of complementarity to their targeted genes, indicated that target transcript cleavage formed the predominant mode of miRNA-directed RNA silencing in this microalga [115,116,117]. However, subsequent reassessment attempts to confidently identify target genes for *Chlamydomonas* miRNAs using the well-established and stringent criteria for miRNA target gene identification in plants, failed to convincingly identify strong candidates as target genes whose expression is regulated by miRNA-directed transcript cleavage [114,118]. What the experimental analysis associated with this bioinformatic assessment did demonstrate though was that for the few miRNAs with a high degree of complementarity to their target transcript(s), mRNA cleavage indeed formed the primary mode of RNA silencing directed by this small subclass of *Chlamydomonas* miRNAs to control target gene expression [114,115,116,117,118]. Furthermore, this revised analysis to target gene identification showed that most *Chlamydomonas* miRNAs only have limited complementarity to their target genes across the seed region of the miRNA, and that putative target sites are frequently positioned in, or proximal to, the 3′ UTR of the target transcript. Given these characteristics shared with animal miRNAs, it is unsurprising that translational repression appears to be the predominant form of RNA silencing directed by this subgroup of miRNAs with only ‘*seed region*’ matches to their target gene transcripts [114,118,119,120,121] (Figure 3). Interestingly, seed region complementarity between a miRNA and its putative targets, led [120] to predict that each *Chlamydomonas* miRNA returned a high enough degree of seed region complementarity to putatively target ~10 protein-coding sequences for expression regulation. Together, these works [114,118,119,120,121] have shown that the cytoplasm localized AGO, *Cre*-AGO3, forms the catalytic core of miRISC in *Chlamydomonas*, and interestingly that *Cre*-AGO3 can direct both a mRNA cleavage and a translational repression mode of RNA silencing to control target gene expression: distinct mechanisms of target gene expression regulation that appear to be determined based on the degree of complementarity of the *Cre*-AGO3-loaded miRNA, to its target gene transcripts.

The terminal nucleotidyltransferase, *Cre*-MUT68, has also been shown to play a core functional role in the *Chlamydomonas* miRNA pathway via controlling the steady state levels of miRNAs and siRNAs (Figure 3). The *Cre*-MUT68 nucleotidyltransferase was originally shown to add untemplated nucleotides to the 5′ cleavage fragments generated by RISC-catalyzed slicing of sRNA target transcripts to identify such cleavage products for decay [122]. Subsequent to the identification of a RISC-generated cleavage fragment clearance role for *Cre*-MUT68, characterization of the *Chlamydomonas mut68* mutant revealed that the levels of numerous miRNAs were elevated in this mutant background along with an elevated abundance of stabilized RISC generated 5′ cleavage fragments [122]. In addition to elevated miRNA levels, *Cre*-AGO3 protein abundance was also elevated in the *mut68* mutant background, which was likely to be the result of a feedback mechanism to attempt to accommodate the increased abundance of the global miRNA population in the *mut68* mutant. Further molecular analysis of the *mut68* mutant, and of the wild-type *Cre*-MUT68 protein itself, showed that with respect to miRNAs, *Cre*-MUT68 functions to preferentially add a tail of U residues to the 3′ terminal nucleotide of miRNAs (i.e., 3′ terminal uridylation), a sequence modification which identifies the U-tailed miRNA for its degradation by the exosome [122,123]. Using an in vitro system, the authors [122] went on to demonstrate that *Cre*-MUT68 could form a functional interaction with the 3′ to 5′ exoribonuclease and exosome subunit protein, Ribosomal RNA processing6 (*Cre*-RRP6) (Figure 3), which in the in vivo setting, would ensure tight regulation of maintenance of miRNA steady state levels via the efficient removal of damaged or unrequired miRNAs in *Chlamydomonas* cells [122]. The quality control surveillance mechanism performed by *Cre*-MUT68 in the *Chlamydomonas* miRNA pathway is highly similar to that performed by the nucleotidyltransferase HEN1 SUPPRESSOR1 (*Ath*-HESO1) in *Arabidopsis* which uridylates unmethylated miRNAs (Figure 1) and siRNAs to tag these RNA fragments for removal from the *Arabidopsis* cell [124]. The terminal uridyltransferase, *Dme*-Tailor, has also been shown to involved in the modification of mirtron pre-miRNAs to control the degree of contribution that the mirtron subclass of miRNAs makes to the global miRNA population of *Drosophila* [125]. Therefore, uridylation of miRNAs, or their precursor transcripts, appears to form a conserved mechanism to add an additional layer of complexity to the miRNA pathways of plants, animals and microalgae [122,123,124,125].

## 5. The microRNA Pathway of Green, Brown and Red Seaweed

### 5.1. microRNAs in the Chlorophyta

Of the Chlorophyta (green algae), the majority of knowledge gained to date on the miRNA pathway has been generated from the molecular characterization of *C. reinhardtii*. The initial miRNA studies in *Chlamydomonas* indicated that well over 100 miRNAs were present in this unicellular green microalga [28,33], with a subsequent study [126] suggesting that over 300 miRNAs were present in *C. reinhardtii*. The actual number of miRNAs in *Chlamydomonas* remains to be experimentally validated, however an excellent bioinformatic study which reassessed the initial predictions for *C. reinhardtii* indicated that the number of bona fide miRNAs in *Chlamydomonas* is likely to be much reduced with many candidates failing to meet the more stringent identification/annotation criteria established for plant miRNAs [35]. Moreover, the [35] reassessment of the [28,33] datasets more accurately reassigned many of the initially identified miRNAs to have originated from a siRNA-generating precursor transcript, and not from a precursor which would be capable of folding to form the dsRNA structure required to be recognized as a processing substrate by *Cre*-DCL3 for miRNA production.

In addition to *C. reinhardtii*, the miRNA profile of the multicellular green microalga, *Volvox carteri* has also been constructed with *V. carteri* traditionally serving as an important species to study the mechanistic basis of the developmental processes of embryonic cleavage, morphogenesis, and cell differentiation [127,128]. Profiling of both the gonidia and somatic cells of *V. carteri* identified ~170 putative miRNA candidates with the abundance of 16 miRNAs experimentally confirmed via the northern blot nucleic acid hybridization approach [129]. Of particular interest was the identification of cell type-specific enrichment of some of the identified *V. carteri* miRNAs in gonidia and somatic cells, and furthermore, the predicted target genes of these cell type-enriched miRNA populations were shown to have an opposing expression trend to the abundance of their potential miRNA regulators in these two developmentally distinct cell types [129]. This finding suggested, that like plant and animal miRNAs, green algae miRNAs potentially mediate key roles in regulating the expression of genes crucial for driving the development of *V. carteri*, and potentially of other microalga species.

In the green microalgae species *Chlorella sorokiniana*, *Coccomyxa subellipsoidea* and *Botryococcus braunii*, 98, 124, and 14 miRNAs, respectively, have been identified [130,131,132]. Interestingly, these analyses have generated differing results with respect to the composition of the individual miRNA populations identified in each microalga, with an abundance of species-specific miRNAs reported in *C. sorokiniana* (i.e., 60 out of 98 miRNAs were unique to *C. sorokiniana*) [130] versus a very high degree of conservation for the miRNAs identified in *C. subellipsoidea* with 118 out of 124 miRNAs detected in *C. subellipsoidea* previously reported for other organisms [131]. Such differences are likely accounted for by (1) the algorithmic basis of the approach used for the initial miRNA identification in each species, together with (2) the degree of stringency of the annotation process post miRNA identification via sequencing. Another interesting finding which stems from the miRNA profiling of these green microalgae is that aside from the *V. carteri* findings [129], target gene assessment in these Chlorophyta microalgae has inferred that the regulatory role directed by miRNAs is highly focused on controlling the expression of genes involved in biological processes such stress responses, chlorophyll production and photosynthesis, lipid, sucrose and starch metabolism, and the biosynthesis of secondary metabolites [130,131,132]. This is in direct contrast to those findings previously reported in both the plant and animal systems where miRNA-directed gene expression regulation has been demonstrated to be an essential molecular component of development. Specifically, a less essential role in controlling developmental gene expression in the green microalgae is most readily demonstrated by the mutational work conducted in *C. reinhardtii* with the *Chlamydomonas dcl3*, *dus16* and *ago3* mutants all displaying largely wild-type-like phenotypes irrespective of the extent of dysfunction of the miRNA pathway [112,113,114]. 

Acting as regulators of gene expression outside of development in algae is further evidenced by miRNAs studies in two other green microalgal species, namely *Dunaliella salina* and *Haematococcus pluvialis* [133,134,135,136]. Due to its high salinity tolerance, *D. salina* is widely distributed globally in salt lakes with *D. salina* accumulating high amounts of β-carotene in plastid-localized lipid globules, which together marks *D. salina* as an ideal model to study the molecular basis of salt tolerance in plants [133], and as a bioresource to produce β-carotene for use in industry or for pharmaceutical applications [134]. A large collection of known and novel miRNAs has been reported in *D. salina* [133,134]. However, although the morphology of *D. salina* changes considerably in an altered growth environment (i.e., high salt or intense light), the majority of the predicted target genes for the miRNAs which changed in their abundance in an altered growth environment where categorized by Kyoto Encyclopedia of Genes and Genomes (KEGG) analysis to function in metabolic pathways, including the metabolism of amino acids, nucleotides, energy, cofactors and vitamins, and to not be involved in regulating cell growth or development [133,134]. In suitable growth environments, the unicellular alga *H. pluvialis* exists in its motile green vegetative form, however, in a nutrient poor environment, *H. pluvialis* transitions into the chlamydospore stage of its lifecycle [135]. In this stage of its lifecycle, *H. pluvialis* losses its flagellum to become nonmotile, its cell wall thickens, and it turns red in color via accumulating a high abundance of the carotenoid molecule, astaxanthin, an important additive to both human and animal feed [135]. Assessment of the miRNA profile of *H. pluvialis* identified 405 miRNAs with homology to known plant miRNAs [135,136], and these were further grouped into 75 *MIR* gene families, including grouping into the highly conserved plant *MIR156*, *MIR159*, *MIR160*, *MIR164* and *MIR167* gene families [135]. As reported for *D. salina* miRNAs, Gene Ontology (GO) and KEGG analyses showed that the putative target genes of *H. pluvialis* miRNAs function in cellular processes such as fatty acid biosynthesis, regulation of ATP and transporter activity, high light and iron stress responses, and most interestingly, regulation of the astaxanthin biosynthesis pathway [135]. The authors [136] went on to show that specific miRNAs are preferentially loaded into the extracellular vesicles (EVs) released by both the motile and nonmotile forms of *H. pluvialis*. Furthermore, many of the EV-enriched miRNAs were subsequently shown to putatively regulate the expression of target genes involved in biological processes relating to EV formation and transport, such as being grouped into the functional categories; cell junction, membrane-enclosed lumen, membrane-part, membrane, organelle part, and organelle [136]. This forms a highly interestingly result as it not only demonstrates the ability of microalgae to form EVs which load specific miRNA cargo, but this finding additionally infers that miRNA-loaded EVs may play a role in cross-kingdom communication between microalgae and their interacting species. Surprisingly, despite the growing volume of work on miRNA biology in Chlorophyta microalgae, to date, no characterization of the miRNA population of a green seaweed species has been reported.

### 5.2. microRNAs in the Phaeophyceae

Like the Chlorophyta, the Phaeophyceae (brown algae) form a large group of photosynthetic organisms, however, via comparison, the brown algae have undergone a much longer and largely independent evolution from their last common ancestor of land plants [137]. Due to its wide global distribution and perceived importance for the study of the evolution of multicellularity in the Phaeophyceae and other seaweeds, the genome of *Ectocarpus siliculosus* was published in 2010 [138], a considerable undertaking which also detailed the presence of homologs of the core pathway machinery proteins *Esi*-DCL1 and *Esi*-AGO1, as well as to report on the identification of 26 miRNAs. The identified miRNA sRNAs, displayed features similar to those of their plant counterparts, with 24/26 *Esi*-miRNAs expressing a 5′ terminal U residue and being 21-nt in length at maturity. At the genome level, the *Esi*-miRNAs also displayed similarities to animal miRNAs, with 17 out of the 26 identified miRNAs determined to have been processed from the intron of a protein-coding host gene [138]. In a subsequent miRNA study of *E. siliculosus*, an extremely large set of over 500 putative miRNA candidates were reported [139], although many of the proposed candidates were shown to vary in length compared to the expected predominant size of mature plant and animal miRNA silencing signals at 21-nt and 22-nt, respectively. However, via quantitative reverse transcriptase polymerase chain reaction (RT-qPCR)-based approaches, the authors did experimentally validate the expression of 22 mature miRNA candidates as well as to confirm the presence of a precursor transcript for 8 of the 22 experimentally validated *Esi*-miRNAs. The RT-qPCR approach was also used to show that the expression of *Esi-DCL1* and *Sci-AGO1* was altered by the cultivation of *E. siliculosus* in a high salt environment, and further, that the abundance of some of the characterized *Esi*-miRNA candidates was also altered accordingly [139]. Via the use of stringent selection criteria, the [138,139] *E. siliculosus* miRNA datasets were reanalyzed by [43], with this analysis firmly establishing that the size of the total miRNA population of *E. siliculosus* is likely composed of ~63 *MIR* gene families. Findings stemming from this study of particular interest included (1) the discovery that only one of the 63 identified *MIR* gene families contained multiple members, (2) the identification of only two *MIR* gene clusters (i.e., individual *MIR* genes within five kilobases (kb) of each other), (3) mapping of the origin of 75% of the miRNAs to protein-coding host gene sequences (44 mapped to introns and 3 mapped to UTRs), and (4) that all experimentally validated miRNAs were at an equivalent level of abundance in both the sporophyte and gametophyte generations of the *E. siliculosus* lifecycle [43]. This final finding adds further weight to the suggestion that miRNAs are not important regulators of the expression of genes crucial to the development of algae.

*Saccharina japonica* (previously *Laminaria japonica*) is a cold temperature species of brown seaweed which is extensively cultivated on ropes for use as a human feed source in the eastern coastal regions of China and Korea as well as western Japan. In the approximate 100-year period that *S. japonica* has been grown in this region its area of cultivation has expanded greatly, including expansion into cultivation zones of temperate and even subtropical waters [140]. The expansion of the growing region of cultivated *S. japonica* cultivars infers that these cultivars have been adapted to growth in warmer waters, a suggestion which led [141] to investigate the contribution which miRNAs may have had at the molecular level to the adaptability of cultivated *S. japonica* to grow at higher temperatures [141]. This research identified 49 conserved and 75 novel miRNAs in *S. japonica*, with the majority of both miRNA classes identified in the control and elevated temperature samples of *S. japonica*. Moreover, 7 conserved and 25 novel *Sja*-miRNAs were determined to significantly differentially accumulate when *S. japonica* was cultivated at a higher growth temperature, suggesting that these 32 miRNAs might represent heat stress responsive miRNAs [141]. A second *S. japonica* study [142] identified a highly similar number of miRNA candidates (n = 117) to the 124 miRNAs identified by [141], and went on to group the 117 miRNAs identified into 98 *MIR* gene families. Of particular surprise was the finding that none of the miRNAs identified in *S. japonica* shared any sequence similarity to the miRNAs previously catalogued for *E. siliculosus* [43,142]. *Saccharina* and *Ectocarpus* separated from their last common ancestor ~95 Mya to show that miRNA populations of brown seaweeds evolve very rapidly. Therefore, this finding added additional weight to the indication that in direct contrast to the plant and animal miRNA systems where core sets of *MIR* gene families are strictly conserved, such an evolutionary requirement is not observed in algae, including both microalgae and the seaweeds. Furthermore, in animal lineages, miRNA evolution from genomic hairpins appears to be the most predominant mechanism of new miRNA generation [143], and considering that none of the *S. japonica* or *E. siliculosus* miRNAs showed any real degree of sequence similarity to repeat sequences or protein-coding genes as has been repeatedly reported in plants [43,142], this also may form the driving mechanism of rapid *MIR* gene evolution in seaweeds.

### 5.3. microRNAs in the Rhodophyta

#### 5.3.1. microRNAs in Red Microalgae

The Rhodophyta (red algae) belongs to the Plantae supergroup and as such forms a large group of micro- and macroalgal photosynthetic organisms due to their common ancestor having hosted a cyanobacterium which subsequently evolved into a plastid via the process of primary endosymbiosis [144,145]. The light-harvesting pigment molecules phycoerythrin and phycocyanin accumulate to high abundance in red algae, with their levels masking the other photosynthesis-related molecules such as the chlorophylls and other predominant carotenoids, a piment molecule profile which gives the Rhodophyta their distinct red color as well as to enable numerous species within this phylum to inhabit diverse environments [146,147,148,149]. Further, numerous Rhodophyta serve as important sources of industrial or medical products due to their biosynthesis of unique biomolecules such as agar and carrageenan gels [146,147,148,149], and this has led to the profiling of the sRNA populations of a restricted number of red micro- and macroalgae to determine the composition of the miRNA landscape in this unique and highly important algae phylum. 

For red microalgae, the miRNA populations of the abundant antioxidant producing species *Porphyridium purpureum*, the industrially promising species *Porphyridium cruentum*, and the extremophilic heterotrophic species *Galdieria sulphuraria* have been reported [150,151,152]. Via a deep sRNA sequencing approach, ref. [150] bioinformatically identified an extensive population of 254 *P. purpureum* miRNAs composed of 203 known and 51 novel miRNAs. Northern blotting and RT-qPCR were used to experimentally validate a select cohort of *Ppu*-miRNAs to provide confidence that a biologically relevant population of miRNAs had been identified for *P. purpureum* [150]. Bioinformatic predictions via GO analysis of *Ppu*-miRNA target genes suggested that most of the putative candidates encode proteins involved in protein transport, signal transduction, and photosynthesis-related processes, including chloroplast membrane production and maintenance, chloroplast relocation, and photoreceptor activity [150]. In the closely related *Porphyridium* spp., *P. cruentum*, 49 miRNAs were identified that were grouped into 46 *MIR* gene families. Interestingly, only two of the identified miRNAs (i.e., *Pcr*-miR156 and *Pcr*-miR419) were reported previously in *P. purpureum* [150], a finding which once again indicates negligeable *MIR* gene conservation in the algae, even between very closely related species [150,151]. However, in spite of the lack of miRNA conservation between these two *Porphyridium* spp., proteins such as Ribulose-1,5-bisphosphate carboxylase oxygenase (RuBisCO: rate limiting enzyme of photosynthesis), light harvesting complex subunits, phycobiliproteins, oxygen-evolving enhancer proteins, and granule-bound starch synthase, all of which perform essential roles in photosynthesis, were also identified in *P. cruentum* as putative miRNA target genes [151], as where many of those putative target genes identified for *P. purpureum* miRNAs [150].

*Galdieria sulphuraria* has evolved to acquire unique biological properties which allow the species to thrive in extreme environments including hot springs, acidic soils and volcanic rocks [153,154]. The miRNA population of *G. sulphuraria* was constructed as a first step towards uncovering a role for miRNA-directed gene expression regulation in the abiotic stress adaptability of this extremophile red microalgae [152]. This endeavor uncovered the presence of 134 *Gsu*-miRNAs, including 120 known and 14 novel miRNAs, with a small subset of those identified experimentally validated via sRNA northern blotting and RT-PCR. Curiously, 21 miRNAs returned homology to members of highly conserved plant *MIR* gene families, whereas no sequence homology was found between *G. sulphuraria* and *C. reinhardtii* miRNAs. Additional characteristics for the *G. sulphararia* miRNA population included that ~92% formed single member *MIR* gene families, a U residue was the predominant 5′ terminal nucleotide, and the miRNAs were predominantly 21-nt in length [152]. The authors went on to reveal that organic substance metabolic processes, chloroplast and thylakoids, electron transport chain, and electron transport, formed the major functional groupings for most putative target genes identified for both the known and novel *Gsu*-miRNA populations [152].

#### 5.3.2. microRNAs in Red Macroalgae

*Porphyra yezoensis* (syn. *Pyropia yezoensis*) is a red seaweed which inhabits the intertidal zone of shallow cold seawaters and is widely cultivated in China, Japan and Korea where it is used to produce sea vegetable-based products, as well as forming a traditional source of edible seaweed in Ireland and Britain [155,156]. Of the ~70 species which belong to the *Porphyra*/*Pyropia* complex worldwide, *P. yezoenesis* is viewed as the best model species owing to its long-established use in biological research and for its agronomic importance [155,156]. *P. yezoensis* has a dimorphic lifecycle composed of two highly morphologically distinct generations, the sporophyte (microscopic diploid form) and gametophyte (macroscopic haploid form) generations [157]. At the molecular level, gene expression analysis via the use of available, yet limited expressed sequence tag (EST) data, showed that less than a quarter of *P. yezoensis* genes are active across its two lifecycle generations [157]. This finding prompted [158,159] to profile the miRNA population of *P. yezoensis* to determine the contribution of miRNA-directed gene expression regulation to the development of two such morphologically distinct forms of this red seaweed. The initial miRNA profiling exercise in 2010 [158] identified 224 candidate *Pye*-miRNAs, including 53 *Pye*-miRNAs which only showed a low level of sequence conservation to known plant miRNAs (and which were subsequently redescribed as *P. yezoensis*-specific miRNAs), 41 *Pye*-miRNAs which matched previously identified *C. reinhardtii* miRNAs (with 17 of these returning high identity scores), 54 *Pye*-miRNAs which are conserved in the ground moss *Physcomitrella patens*, and 16 *Pye*-miRNAs with no sequence identity to any other previously reported miRNA isolated in either plants or algae (i.e., classed as novel or species-specific *Pye*-miRNAs). The target sequences of the identified *Pye*-miRNAs were found to be primary located in the coding region of putative target genes with identified targets assigned to a wide range of biological functions, including chromatin structure and dynamics, translation – ribosome structure and biogenesis, and signal transduction. Interestingly, the genes putatively targeted for expression regulation by the 16 novel miRNAs identified were determined to largely encode proteins of no known molecular function, a finding that tentatively suggests that these target genes may perform functions specific to the red seaweeds, or even to *P. yezoensis* itself [158]. 

In direct contrast to this initial study, a subsequent *P. yezoensis* miRNA study, which applied the much more stringent rules used for miRNA identification in plants, only identified 14 *Pye*-miRNA candidates [159]. Further, the authors [159] showed that the pre-miRNA precursor transcripts of the 14 identified *Pye*-miRNAs shared structural properties highly similar to those previously reported for well-characterized plant miRNA precursor transcripts. In addition, this work revealed that only a single *Pye*-miRNA was expressed during both the sporophyte and gametophyte generations of the *P. yezoensis* lifecycle, while seven and six *Pye*-miRNAs were exclusively expressed in the sporophyte and gametophyte generation, respectively [159]. This finding infers that these 13 *Pye*-miRNAs may mediate generation-specific molecular processes in *P. yezoensis*. Unfortunately, the absence of genome sequence data, and molecular resources (e.g., the availability of extensive EST libraries), resulted in the failure of [159] to confidently predict the likely biological functions of most identified putative target genes for the 14 *Pye*-miRNAs isolated in their study.

*Chondrus crispus* is an intertidal red seaweed which naturally inhabits the rocky shorelines of the north Atlantic coast, and like other red seaweeds, *C. crispus* has a complex cell wall composed of cellulose, hemicelluloses, and unique sulphated galactans such as carrageenan [160,161]. Such red algal-specific molecules have rocketed numerous Rhodophyta species into the research spotlight, including *C. crispus*, considering that they have wide industrial applications as well as to serve as an important food source for humans and livestock: uses worth hundreds of millions of dollars annually [162,163]. Sequencing and annotation of the ~105 megabase (Mb) *C. crispus* genome revealed that most genes are of small size being composed of a single exon (monoexonic), and of those protein-coding sequences which harbor introns, the introns are widely dispersed and are of a short length [164]. It is interesting to note that those genes with introns are expressed at a higher level than monoexonic genes, a finding which indicates an important regulatory function for intronic sequences in *Chondrus*. Although most genes are of small size and lack introns, *C. crispus* was revealed to encode both a *Ccr*-DCL and *Ccr*-AGO endonuclease to tentatively suggest that sRNA silencing pathways are functional in this red seaweed [164]. In addition to encoding these core machinery proteins of RNA silencing, the *C. crispus* nuclear genome was revealed to harbor a large, complex, and functionally diverse set of genes involved in halogen metabolism related processes, including heme peroxidases, phosphatidic acid phosphatases, halo peroxidases, haloalkane dehalogenases, and haloacid dehalogenases [164]. These genome-based findings identified *C. crispus* as an excellent model to study miRNA-directed molecular control of halide chemistry and halogen metabolism. 

The combined use of sRNA sequencing together with the bioinformatic analysis of available *C. crispus* EST data, enabled the identification of 117 *Ccr*-miRNAs, including 108 known miRNAs and nine novel *Ccr*-miRNAs [165]. These 117 miRNAs were subsequently grouped into 110 *MIR* gene families, including 103 single member families and five, one and one *MIR* gene families which included two, three and four members, respectively. In addition, the most frequent length of *C. crispus* miRNAs upon maturity was 19-nt, and not 21-nt like plant miRNAs to which many of the known *Ccr*-miRNA displayed sequence homology [165]. Northern blot hybridization (n = 3) and RT-qPCR (n = 18) were also used to experimentally validate 21 randomly selected *Ccr*-miRNA sRNAs. The authors used plant miRNA targeting requirements of near perfect complementarity between the targeting miRNA and the target site of the regulated mRNA to identify 160 putative target genes for the 117 *Ccr*-miRNAs. Target gene GO analysis indicated that most putative targets function in nucleotide binding, protein folding, and intracellular membrane-bound organelle processes. Of particular interest was the finding that via the use of KEGG mapping, 22 genes identified as putative targets of *Ccr*-miRNA-directed expression regulation, perform functional roles in peroxisome proliferator-activated receptor (PPAR) signaling pathways [165]. This finding strongly suggests that such signaling pathways require exquisite regulatory control in *C. crispus*, and potentially in other red seaweed species.

Together, Shulian Xie’s group have identified 469 *MIR* gene families, composed of 503 members, across three representative red algal species, including the red microalgae *P. purpureum* [150] and *G. suphuraria* [152], and the red seaweed *C. crispus* [165]. Via comparative analysis of the three miRNA datasets generated by this research group, 64 miRNA homologs were identified between *P. purpureum* and *G. suphuraria*. In addition, this analysis showed that 72 and 186 *MIR* gene family representatives were conserved between *C. crispus* and *P. purpureum*, and *C. crispus* and *G. suphuraria*, respectively [150,152]. Many of the miRNAs conserved in these red algal species were originally identified in plants, including some of the most highly conserved *MIR* gene families in plants, such as the *MIR156*, *MIR159*, *MIR164*, *MIR165*/*166*, *MIR168*, *MIR169*, *MIR396* and *MIR398* families. The authorship team then went on to use the miR156 sRNA sequence for further analysis and showed that the sequences of *Ppu*-miR156, *Gsu*-miR156 and *Ccr*-miR156, retained an approximate 90% identity to the 294 mature miR156 sequences entered into miRBase from 47 plant species. This analysis also showed that although the *PRE-MIR156* precursor transcripts had diverged widely in plants, and the three assessed red algal species, the mature miR156 sequence has remained highly conserved to indicate that miR156 controls the expression of genes which perform an essential function in both plants and red algae [152]. In *Arabidopsis*, the *MIR156* gene family contains ten members (*Ath*-miR156a to *Ath*-miR156j) with family members regulating the expression of a small subclade of the *SQUAMOSA PROMOTER BINDING PROTEIN-LIKE* (*SPL*) transcription factor gene family, including *SPL9* and *SPL10*, which function together to initiate the transition of *Arabidopsis* from vegetative to reproductive development [1,2]. In *C. crispus* however, the predicted target gene of *Ccr*-miR156, *CHC_T00006037001*, encodes a putative arsenite/tail-anchored protein-transporting ATPase protein possibly involved in arsenite transport across cell membranes for detoxification [152]. This finding therefore raises the intriguing question of why would the mature miR156 sequence be so tightly conserved between plants and red algae when miR156 regulates the expression of such functionally distinct target genes?

*Eucheuma denticulatum*, commonly known as “Spinosum”, is an important cultivated red seaweed species in warmer tropical waters forming an important source of carrageenan in the food industry and dietary fiber for blood health [166,167,168]. In another study from Shulian Xie’s group, 133 miRNAs were identified for *E. denticulatum*, which included homologs of members of 21 highly conserved plant *MIR* gene families such as the *MIR164*, *MIR166*, *MIR169*, *MIR397* and *MIR398* gene families [169]. The authors went on to experimentally validate three and 13 *Ede*-miRNAs via sRNA-specific northern blot hybridization and RT-qPCR, respectively. Considering the high degree of conservation of the identified *Ede*-miRNAs with plant miRNAs, plant parameters were applied to putatively identify 871 targets for the 133 *Ede*-miRNAs with ~70% of identified targets determined to likely be under transcript cleavage-based expression regulation, and the remaining ~30% possibly forming targets for the translation repression mode of miRNA-directed RNA silencing [169]. Furthermore, KEGG analysis revealed that the most significantly enriched pathways for the identified putative target genes were biosynthesis of secondary metabolites and metabolic pathways, including the enrichment of both the porphyrin and chlorophyll metabolism pathways.

*Gracilaria lemaneiformis* (syn. *Gracilariopsis lemaneiformis*) is also a widely cultivated red seaweed due to its high yield potential and is used to produce a considerable volume of agar and other commercially important polysaccharides and as a feed source in aquaculture [170,171]. As part of a large-scale transcriptome analysis of *G. lemaneiformis*, ref. [172] identified 91 miRNAs with 23, 28 and 40 *Gla*-miRNAs returning homology to known plant, microorganism and animal miRNAs, respectively. The growth and development, and therefore yield potential of *G. lemaneiformis* is adversely affected by higher seawater temperatures, and by heavy metal pollutants [173,174]. The expression of the precursors of three *Gla*-miRNAs with homology to mammalian miRNAs was therefore assessed in *G. lemaneiformis* samples cultivated at either a high temperature or in the presence of heavy metals. However, the expression of the pre-miRNAs of the three analyzed miRNAs showed little variation across the samples cultivated under different growth conditions [172]. Therefore, the extent of the role of miRNAs in the regulating the growth and development of *G. lemaneiformis* in less favorable environmental conditions could not be ascertained by this work. 

In addition to *G. lemaneiformis*, the miRNA profile of a second *Gracilaria* species., *G. vermiculophylla* has been established [175]. *Gracilaria vermiculophylla* (syn. *Gracilariopsis vermiculophylla*) is a red seaweed species native to the coasts of the northwest Pacific Ocean, but over the last approximate 100 years, this seaweed has spread (via its unintentional introduction) to growing regions in the northwestern coastlines of the African continent, as well as to European coasts and North American shores [176,177] where it has thrived due to its enhanced tolerance to environmental stress and being less palatable to the herbivores native to its introduced areas of growth [178,179]. Profiling of the miRNA populations of *G. vermiculophylla* tetrasporophytes sampled from geographically distinct locations (i.e., the east coasts of North America and Japan and the northwest coast of Germany), established a large population of both known and novel miRNAs, many of which were also shown to have differing abundance levels in each assessed *G*. *vermiculophylla* tetrasporophyte sample [175]. In addition, for a number of the differentially accumulating miRNAs, some of the predicted gene targets displayed expression trends which opposed those of their potential miRNA regulator, sRNA and mRNA abundance trends which tentatively indicate that miRNA-directed target gene expression regulation may play a role in the adaptability of *G*. *vermiculophylla* to flourish in diverse aquatic environments.

#### 5.3.3. The miRNA Pathway of the Red Seaweed *Asparagopsis taxiformis*

*Asparagopsis taxiformis* is a red seaweed species which is widely distributed globally in warm to tropical seawaters [180,181,182] and has garnered considerable attention in recent years stemming from its synthesis of biomolecules of unique chemical composition for use in agriculture, or the food technology and pharmaceutical industries [183,184,185,186,187]. Namely, *A. taxiformis* is unique in its ability to not only synthesize the halogenated organic compound bromoform, but to store bromoform at high quantities in specialized gland cells which can constitute up to 4.0% of the total dry weight of *A. taxiformis* [188]. Bromoform has antimicrobial activity and can alter the gut microbiome of ruminant livestock [189,190,191]. Specifically, when *A. taxiformis* is included in ruminant feedstock at levels as low as 2.0% of feed dry matter, the bromoform provided by *A. taxiformis* supplementation can reduce methane production by ruminants to almost undetectable levels [189,190,191]. Considering their demonstrated role as master regulators of gene expression, miRNA research should form a central component of the future molecular characterization of *A. taxiformis*, for this emerging commercial crop with industry investments across the Indo-Pacific region. A draft genome is available for *A. taxiformis*, and therefore, miRNA investigations would provide researchers and industry with an additional and novel avenue to gain a full understanding of all genetic components of the bromoform and other natural product biosynthesis pathways of this red seaweed species. As a first step towards achieving this goal, we have recently started to interrogate our genomic [192] resources to attempt to establish the extent to which the miRNA biosynthesis pathway exists in *A. taxiformis*. Via this approach, we have identified at high confidence, direct homologs of the core protein machinery of the *Arabidopsis*, *Drosophila* and *Chlamydomonas* miRNA pathways. Specifically, Figure 4A provides a schematic representation of the proposed miRNA pathway for *A. taxiformis* based on the pieces of homologous protein machinery uncovered by the bioinformatic interrogation of our *A. taxiformis* genomic data. In addition, Figure 4B details the functional domain landscape of each piece of miRNA pathway protein machinery identified as part of our bioinformatic analyses. The full amino acid sequence of the coding sequences of each piece of protein machinery core to the *A. taxiformis* miRNA pathway is also provided in Appendix A.

Interestingly, although this approach readily identified direct *A. taxiformis* homologs of the protein machinery essential for miRNA precursor transcript processing in the nucleus of eukaryotic cells, *Ata*-DCL3 and *Ata*-SE1 (homologs of *Ath*-DCL1/*Dme*-Drosha/*Cre*-DCL3 and *Ath*-SE1/*Dme*-Ars2, respectively), we failed to identify any candidate gene which could encode for a homolog of the DRB proteins, *Ath*-DRB1, *Dme*-Pasha and *Cre*-DUS16 [64,86,87,112,113]. In *Arabidopsis*, *Drosophila* and *Chlamydomonas*, *Ath*-DRB1, *Dme*-Pasha and *Cre*-DUS16, have been demonstrated to form functional partnerships with the Dcr endonucleases *Ath*-DCL1, *Dme*-Drosha and *Cre*-DCL3, respectively [65,86,113]. Once formed, these protein partnerships ensure accurate and efficient miRNA production right at the very commencement of their respective miRNA biogenesis pathways. The absence of a DRB homolog suggests that at this stage of the *A. taxiformis* miRNA pathway, it operates most analogously to the *Chlamydomonas* miRNA pathway where *Cre*-DUS16 appears to solely provide functional support to *Cre*-DCL3 for miRNA precursor transcript processing [112,113]. This would be in direct contrast to the functional processes operating at this stage of the plant and animal miRNA pathways with *Ath*-DCL1 requiring the assistance of both *Ath*-SE1 and *Ath*-DRB1 for miRNA precursor transcript processing in *Arabidopsis* [61,62,63,64,65,68,70], and the functional interaction requirement of *Dme*-Drosha with *Dme*-Ars2 and *Dme*-Pasha for pri-miRNA processing in *Drosophila* [84,85,86,87,88]. We also failed to uncover an *Ath*-HEN1 homolog in our datasets. However, this was not unexpected as the 2′-O-methylation marking of sRNAs as part of their functional maturation is largely plant-specific [69,70], with neither *Drosophila* nor *Chlamydomonas* miRNAs being modified at their 3′ terminal nucleotide during their production.

In *Drosophila*, *Dme*-Drosha/*Dme*-Pasha processed pre-miRNAs are exported from the nucleus to the cytoplasm via the action of *Dme*-XPO5, a Ran-GTP-dependent dsRNA-binding protein [89,90], with *Ath*-HST performing a similar nucleus exportation process of mature miRNA sRNAs in *Arabidopsis* post processing from their precursors by the microprocessor complex (i.e., *Ath*-DCL1/*Ath*-DRB1/*Ath*-SE1) [72,73]. *A. taxiformis* was determined to encode both a *Dme*-Ran-GTP and a *Dme*-XPO5 homolog. Interestingly, no direct *Dme*-XPO5/*Ath*-HST homolog has been reported for *Chlamydomonas* to suggest that in the *A. taxiformis* miRNA pathway, the nucleus to cytoplasm shuttling of pre-miRNAs or mature miRNAs operates similarly to that of the *Drosophila* and *Arabidopsis* pathways. Our analyses also uncovered *Ata*-AGO3, an AGO protein with close homology to *Chlamydomonas* AGO3 [114,120,121]. However, to date, due to a lack of miRNA data for *A. taxiformis*, together with considerable conjecture as to whether AGO-loaded miRNAs direct either a transcript cleavage or translational repression mode of RNA silencing for target gene expression control in the algae [114,115,116,117,118,119,120,121], future detailed molecular assessment is required to determine whether *Ata*-AGO3 is a cleavage-competent AGO like *Ath*-AGO1. Finally, our bioinformatic interrogation of our *A. taxiformis* genomic data identified an *A. taxiformis* protein with extensive homology to *Ath*-HESO1 and *Cre*-MUT68 [122,123,124]. Moreover, the identified candidate protein, ATA06450, was named *Ata*-HESO1 considering that it returned a much higher degree of identity to *Ath*-HESO1 than to *Cre*-MUT68 [122,123,124]. Identification of *Ata*-HESO1 indicates that this *A. taxiformis* candidate protein putatively functions as a nucleotidyltransferase to uridylate either unmethylated miRNAs, or their precursor transcripts (like the action of *Dme*-Tailor as part of the production of mirtrons in *Drosophila*) to fine tune the rate of miRNA turnover. Considering this promising start to characterization of the *A. taxiformis* miRNA pathway, the next step must be to construct the profile of the global miRNA population of *A. taxiformis* to further compare and/or contrast this essential gene regulatory pathway in this rapidly emerging model red seaweed species to those well-established miRNA pathways in plants, animals and microalgae.

## 6. Conclusions and Future Perspectives

Profiling of the global miRNA populations of micro- and macroalgal species has revealed a considerable lack of *MIR* gene conservation with members of the same *MIR* gene family failing to be isolated in even closely related species which belong to the same phylum [43,142,150,151]. A comprehensive lack of *MIR* gene conservation in the algae is further evidenced via the repeated reporting that most algal *MIR* gene families are single member families [28,33,43,152,165], a finding that is indicative of rapid *MIR* gene evolution in the algae. There is also the requirement to reassess the miRNA profiles of those macroalgal species where considerable conservation to plant and microalgal miRNAs has been reported, and especially for those species where conservation to animal miRNA sequences has also been reported [150,152,158,165,192]. This will ensure that only bona fide miRNAs are identified. The [35] study provides an excellent example of this requirement. Namely, the initial assessment of *Chlamydomonas* indicated that the miRNA profile of this green unicellular alga was composed of well over 100 individual miRNAs [28,33,126]. However, the reanalysis of these datasets via the use of the more stringent criteria established for plant miRNA identification and/or annotation [35], greatly reduced the total number of *Chlamydomonas* miRNAs to a population of just over 60 *MIR* gene families. Similarly, the initial analysis of *E. siliculosus* suggested a miRNA population of over 500 unique sRNAs [139], however, the reanalysis of this data where the highly stringent set of criteria established for plant miRNA identification was applied, a global miRNA population composed of ~63 *MIR* gene families was recalculated for *E. siliculosus* [43]. Due to the variable length of plant miRNA precursors, a defined set of required characteristics have been proposed [193] for the confident identification of miRNAs in plants, including (1) the miRNA guide strand must be processed from the opposite stem arm to the miRNA* passenger strand, (2) the miRNA and miRNA* sequences must hybridize to each other with 2-nt overhangs present at the 3′ terminus of each duplex strand, (3) four or less mismatched base pairings between the miRNA and miRNA* strand of the hybridized duplex, (4) a small number of asymmetric bulges to be present in the stem-loop structured precursor with each composed of one or two nucleotides only, and (5) a stem-loop structured dsRNA can readily fold to form between the miRNA guide strand and miRNA* passenger strand [193]. The application of these criteria will allow for the confident identification of likely miRNA candidates in macroalgal species such the red seaweed *A. taxiformis* for which miRNA data currently does not exist.

In contrast to plant and animal miRNAs, functional predictions for the putative target genes of algal miRNAs suggest that many targets direct roles outside of development [150,151,152,158,159,165]. A minor to negligible role in regulating developmental processes for algal miRNAs is best evidenced by the wild-type-like phenotypes displayed by the *C. reinhardtii dcl3*, *dus16* and *ago3* mutants: mutants defective in the functional activity of the core protein machinery of the *Chlamydomonas* miRNA pathway, *Cre*-DCL3, *Cre*-DUS16 and *Cre*-AGO3, respectively [112,113,114]. Similarly, in the brown seaweed *E. siliculosus*, all experimentally validated miRNAs were shown to accumulate to a similar degree of abundance in both the sporophyte and gametophyte generations of the *E. siliculosus* lifecycle, a finding which indicates that most *Esi*-miRNAs regulate the expression of target genes that perform biological functions distinct to development [43]. Together, these findings infer that the miRNA population of each algal species under analysis should be determined to uncover the species-specific functions controlled by miRNA-directed gene expression regulation. Uncovering the miRNA population of each specific species of seaweed under assessment is also required as the basis of development of highly efficient miRNA technologies such as the development of amiRNA technology. The amiRNA technology has proven to be a highly useful tool in *Arabidopsis* and in other plant species for the further characterization of miRNA function, to provide resistance to invading pathogens, or for the manipulation of the biosynthesis pathways of agronomically important biomolecules [194,195,196,197]. The molecular manipulation of components of the miRNA pathway for miRNA biology research or to alter biomolecule biosynthesis pathways has also been successfully achieved in *Chlamydomonas* via the use of amiRNA technology [111,115,116,117]. Therefore, for seaweed species such as *A. taxiformis* where biological research of the miRNA pathway is only in its infancy, the requirement to establish the profile of the miRNA population of *A. taxiformis* for the future development of miRNA-based technologies is readily apparent. Namely, a highly abundant endogenous *Ata*-miRNA, one which is processed from a structurally simple precursor transcript, could be used as the basis to develop amiRNA technology to manipulate key rate limiting steps of the bromoform biosynthesis pathway to alter the levels of this agriculturally important biomolecule as part of the development of *A. taxiformis* as a model species for future biotechnological research on red seaweeds.

Finally, based on the data discussed in this review, we propose a practical workflow for the reliable large-scale identification and functional characterization of miRNAs in seaweeds, as summarized in the flowchart presented in Figure 5. Given the unique biological and ecological characteristics of seaweeds, miRNA research in this group of species requires careful consideration of sample collection strategies, sequencing quality control, bioinformatics analysis, and functional validation. Our workflow begins with comprehensive sample collection, ensuring the inclusion of diverse developmental stages and environmental conditions to minimize sampling bias. For seaweed species where a nuclear genome sequence is available, or where transcriptomic data exists, a bioinformatic approach should next be applied to identify core miRNA pathway machinery proteins such as the Dcr/DCL and AGO endonucleases. This initial bioinformatic-based analysis can be extended to ascertain the presence of essential functional domains within the identified pieces of core protein machinery, as well as to determine domain structure and composition, with the collective results of such analyses used to predict whether the miRNA pathway of the specific seaweed species under analysis in more functionally similar to the well-characterized pathways of plants, animals or microalgae. Comparing seaweed miRNA pathways to plant- and animal-like mechanisms would provide insight into the mode of action directed by individual members of the miRNA population, such as whether miRNA target gene expression is likely controlled via either a mRNA cleavage or translational repression mode of RNA silencing.

Post core protein machinery identification, the generation of a high-quality sRNA sequencing library is essential. Then, bioinformatics is again required to apply rigorous filtering to remove low-quality reads and contaminants, as well as to ensure a sufficient sequencing depth to capture the typically low abundance of miRNAs. The bioinformatic miRNA identification process integrates structural predictions, conservation analysis, and verification of miRNA/miRNA* duplex formation to improve the accuracy of the process, while in parallel, accounting for the lower conservation levels observed for algal miRNAs. This step must also discard sRNAs originating from siRNA-generating loci, to reduce or even eliminate the likelihood of incorrectly identifying siRNAs as newly identified or species-specific miRNAs. Finally, the workflow extends to application development, including the design of amiRNA tools tailored to seaweed biology, with potential applications in aquaculture and biotechnology. This structured, stepwise approach offers a robust foundation for future research, facilitating high-confidence miRNA discovery and functional validation studies in non-model algal species. By establishing standardized methodologies, our proposed workflow enhances the reliability and reproducibility of miRNA research in seaweeds, paving the way for novel insights into regulatory networks and the potential development of novel biotechnological applications.

## Figures and Tables

**Figure 2 genes-16-00442-f002:**
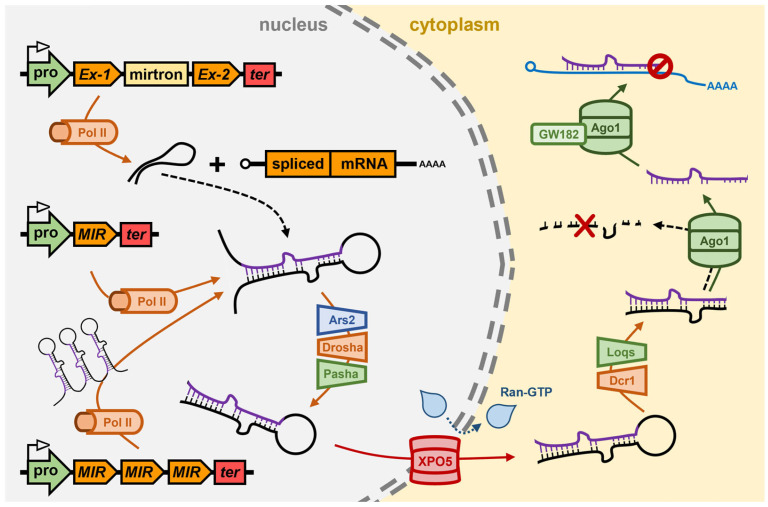
The miRNA pathway of the model animal *Drosophila melanogaster*. In the nucleus of the *Drosophila* cell, Pol II transcribes the pri-miRNA from a *MIR* gene. Pol II is also responsible for the transcription of mirtrons from host genes or for the transcription of *MIR* gene clusters. After folding into a stem-loop dsRNA structure, the pri-miRNA is bound by Ars2 and is then processed into a pre-miRNA by Drosha/Pasha. The pre-miRNA is exported from the nucleus to the cytoplasm by the RanGTP-dependent exportin, XPO5. In the cytoplasm of the *Drosophila* cell, the pre-miRNA is further processed into the miRNA/miRNA* duplex by Dcr1/Loqs. After duplex strand separation and degradation of the miRNA* passenger strand, Ago1 uses the loaded miRNA to guide the silencing of target gene transcripts via translational repression, a mode of miRNA-directed gene expression regulation that also requires GW182. Ago1, Argonaute1; Ars2, Arsenite resistance protein2; Dcr1, Dicer1; GW182, glycine/tryptophan repeat protein 182; Loqs, Loquacious; *MIR*, microRNA gene; mirtron, miRNA precursor-containing intron; Pol II, RNA polymerase II; pro, *MIR* gene promoter; Ran-GTP, RAS-related nuclear protein GTPase; ter, *MIR* gene terminator; XPO5, Exportin-5.

**Figure 3 genes-16-00442-f003:**
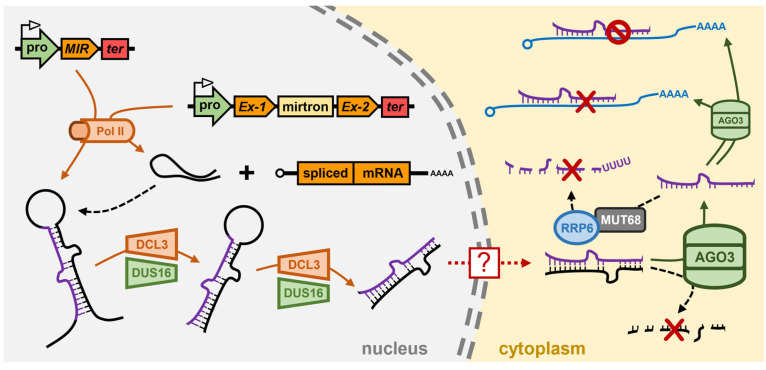
The miRNA pathway of the model green microalgae *Chlamydomonas reinhardtii*. In the nucleus of a *Chlamydomonas* cell, Pol II transcribes pri-miRNAs directly from *MIR* genes, or from protein-coding host genes with the mirtron pri-miRNA forming post intron splicing. Post folding into a stem-loop dsRNA structure, the pri-miRNA is processed into a pre-miRNA by the DCL3/DUS16 partnership. DCL3/DUS16 are also required for further processing of the pre-miRNA into the miRNA/miRNA* duplex to indicate that all processing steps of the production stage of the *Chlamydomonas* miRNA pathway occur in the nucleus. The protein mediator of the export of the miRNA/miRNA* duplex out of the nucleus and into the cytoplasm of a *Chlamydomonas* cell remains unknown (red boxed question mark). In the cytoplasm, the miRNA/miRNA* duplex strands are separated from each other by an unknown mechanism, and then the miRNA* passenger strand is degraded. AGO3 retains the miRNA guide strand and uses it to direct expression regulation of target gene transcripts via either a mRNA cleavage or translational repression mode of RNA silencing. The steady state levels of *Chlamydomonas* miRNAs are controlled by MUT68, together with RRP6, to add an additional layer of regulatory complexity to miRNA-directed gene expression control in *Chlamydomonas*. AGO3, Argonaute3; DCL3, Dicer-like3; DUS16, Dull slicer-16; *MIR*, *MICRORNA* gene; mirtron, miRNA precursor containing intron; MUT68, mutant 68 (terminal nucleotidyltransferase); Pol II, RNA polymerase II; pro, *MIR* gene promoter; RRP6, Ribosomal RNA processing6; ter, *MIR* gene terminator; ?, unknown protein factor.

**Figure 4 genes-16-00442-f004:**
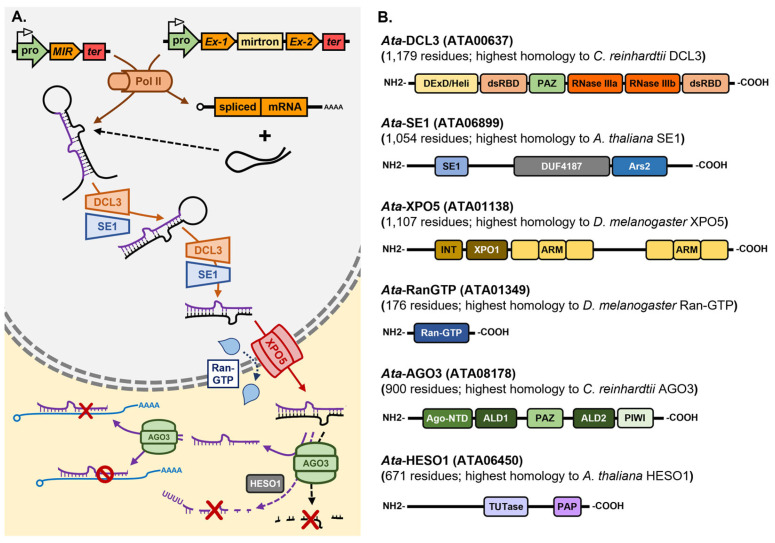
Proposed miRNA pathway for the red seaweed *Asparagopsis taxiformis*. (**A**) Pol II transcribes pri-miRNAs from *MIR* genes or mirtron precursors from protein-coding host genes. DCL3, with the assistance of the RNA-binding protein SE1, processes pri-miRNAs into pre-miRNAs and then pre-miRNAs into miRNA/miRNA* duplexes in the nucleus of *A. taxiformis* cells. The miRNA/miRNA* duplex is exported from the nucleus to the cytoplasm of the *A. taxiformis* cell by a Ran-GTP-dependent XPO5-mediated process and is loaded into AGO3. After the removal and degradation of the miRNA* passenger strand, AGO3 uses the loaded miRNA as a sequence specificity guide to direct the silencing of miRNA target genes via either mRNA cleavage or the translational repression mode of gene expression control. The steady-state levels of miRNAs are controlled in the *A. taxiformis* cell cytoplasm by the activity of the nucleotidyltransferase, HESO1. AGO3, ARGONAUTE3; DCL3, DICER-LIKE3; HESO1, HEN1 SUPPRESSOR1; *MIR*, *MICRORNA* gene; mirtron, miRNA precursor-containing intron; Pol II, RNA polymerase II; pro, *MIR* gene promoter; ter, *MIR* gene terminator; Ran-GTP, RAS-related nuclear protein GTPase; SE1, SERRATE1; XPO5, EXPORTIN-5. (**B**) Schematic representation of the functional domain landscape of *Asparagopsis* homologs of the core machinery proteins of the *Arabidopsis*, *Drosophila*, and *Chlamydomonas* miRNA pathways including *Ata*-DCL3, *Ata*-SE1, *Ata*-XPO5, *Ata*-RanGTP, *Ata*-AGO3, and *Ata*-HESO1. Ago-NTD, Argonaute N-terminal domain; ALD, Argonaute linker domain; ARM, Armadillo repeat domain; Ars2, Arsenite-resistance2 domain; DExD/Hel, DExD/H-box helicase domain; DUF4187, domain of unknown function 4187; dsRBD, dsRNA-binding domain; INT, Importin-β N-terminal domain; PAP, nucleotide poly A polymerase domain; PAZ, Piwi, Argonaute, and Zwille domain; PIWI, Piwi domain; Ran-GTP, Ran small GTPase domain; RNase III, Ribonuclease III domain; SE1, Serrate1/Ars2 N-terminal domain; TUTase, nucleotidyltransferase domain; XPO1, Exportin1-like domain.

**Figure 5 genes-16-00442-f005:**
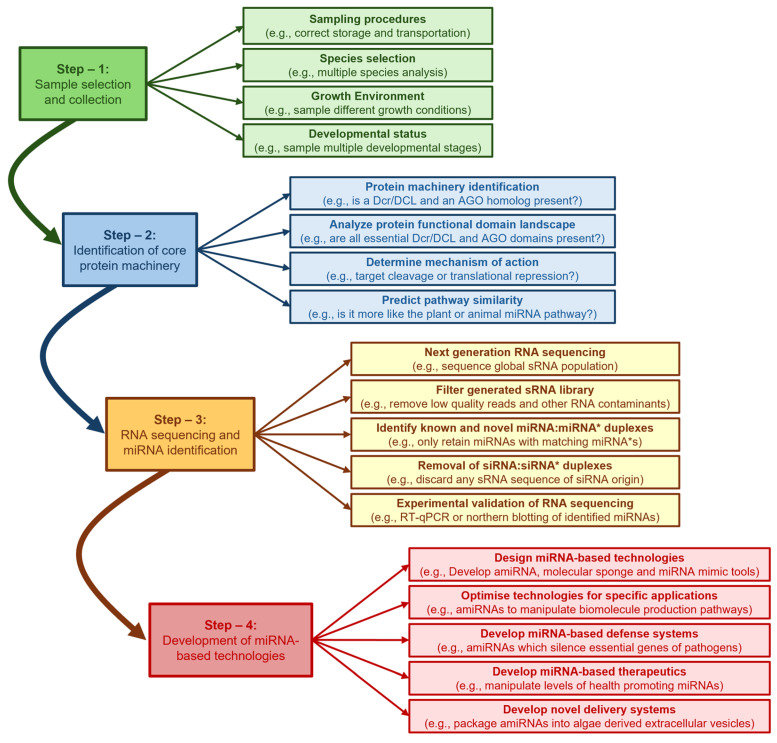
Proposed best practice approach to miRNA identification, characterization, and technology development in seaweeds. The flowchart outlines the four key steps for identifying, analyzing and validating miRNAs in seaweeds. The process is divided into four major modules, which include (1) sample collection, (2) identification of core protein machinery and the inference of miRNA-directed regulatory mechanisms, (3) sRNA sequencing for sRNA library generation and miRNA identification and annotation via bioinformatics, together with experimental validation, and (4) application development. Quality control measures are emphasized at each step to ensure reliability and accuracy in miRNA discovery and functional validation.

## Data Availability

All data discussed in this review are in the public domain.

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
