# Peer review of "The microRNA Pathway of Macroalgae: Its Similarities and Differences to the Plant and Animal microRNA Pathways"

_genes, 2025, doi:10.3390/genes16040442_

Round 1
Reviewer 1 Report
Comments and Suggestions for Authors
The article by Jessica Webb entitled “The microRNA pathway of macroalgae: its similarities and differences to the plant and animal microRNA pathway” is review summarising literature knowledge. The authors compare and contrast the macroalgae miRNA pathway to those well-characterized pathways of plants and animals. The authors have to address some points.
My concerns are:
- The authors have to introduce under 1. first of all the different classes of small regulatory RNAs. The authors must mention all small regulatory RNAs like small Cajal body RNAs, small nuclear RNAs, small nucleolar RNAs, sno-derived RNAs, vault RNAs and the other small RNAs present in organisms.
- An abbreviation should only be introduced when it is also used later in the text. But if an abbreviation has been introduced, the authors have to use it in the following text. The authors have to check the manuscript in regard to these points. Furthermore, they have to summarise all abbreviations in an abbreviations list.
- Sometimes citations for statements are missing; the authors have to prove all statements with citations; e.g. lines 35, 42, 145 (these are only three examples and citations are missing more often). Sometimes even more than one citation is necessary, e.g. line 44 (…for cancer in humans…).
- sRNA-directed gene expression regulation in regard to chronic kidney disease is not only limited to domesticated dogs, it is also proven in humans (DOI 10.1038/bjc.2016.329).
- The statement in line 72 is not correct. MiRNAs can also activate gene expression – one example is DOI 10.1007/978-981-10-4310-9_6.
- The authors have to explain why they are focusing on three model species (lines148-150).
- All abbreviations used in the figures must be explained below in the relevant figure legends.
- It is neither nice nor necessary to read “…. Molnar et al. .…”, “…Zhao et al.” etc. If somebody is interested in the name of the first author (s)he will find this information in the Reference list. The authors have to rephrase all these parts.
- Size of Figure 5 must be increased.
Author Response
Responses to Reviewer 1 Comments.
Comment 1.
The authors have to introduce under 1. first of all the different classes of small regulatory RNAs. The authors must mention all small regulatory RNAs like small Cajal body RNAs, small nuclear RNAs, small nucleolar RNAs, sno-derived RNAs, vault RNAs and the other small RNAs present in organisms.
Response 1.
Our review is exclusively focused on the microRNA class of small regulatory RNA. We have clearly outlined that microRNAs and small-interfering RNAs form the two main classes of small regulatory RNAs. We therefore are of the opinion that the addition of text discussing all of the other small-interfering RNAs species which have been reported in only specific clades of the eukaryotes would only distract the reader’s attention from our primary focus on microRNAs. We have however added a statement in the Acknowledgements section of the review detailing that we could not cover all aspects of small RNA biology in the review.
Comment 2.
An abbreviation should only be introduced when it is also used later in the text. But if an abbreviation has been introduced, the authors have to use it in the following text. The authors have to check the manuscript in regard to these points. Furthermore, they have to summarise all abbreviations in an abbreviations list.
Response 2.
We thank Reviewer 1 for identifying this oversight. We have revised the manuscript accordingly to address this issue. We would like to highlight however that there are instances where we have again provided the full species name when pages of text separate the first, and subsequent mention of a specific species. We believe that this is approach is helpful to the reader. The only other remaining instance where we use the full name post stating an abbreviation is for nucleotide (nt). We use ‘nt’ when discussing the length of a transcript, whereas the use of ‘nucleotide’ is more appropriate when discussing a single or specific residue.
Again, we thank Reviewer 1 for this excellent suggestion. We apologise for this oversight. An abbreviation list has been included in the revised manuscript.
Comment 3.
Sometimes citations for statements are missing; the authors have to prove all statements with citations; e.g. lines 35, 42, 145 (these are only three examples and citations are missing more often). Sometimes even more than one citation is necessary, e.g. line 44 (…for cancer in humans…).
Response 3.
All factual statements which require citation has been referenced accordingly. Furthermore, we have referenced approximately 200 previous studies in this review, therefore, supplying multiple references for specific statements is not required. The authors are of the opinion that a reference has been supplied on every occasion where appropriate throughout the manuscript. As outlined in Response 1, we have included a statement in the Acknowledgements section of the review detailing that we could not cover all aspects of small RNA biology in the review.
Comment 4.
sRNA-directed gene expression regulation in regard to chronic kidney disease is not only limited to domesticated dogs, it is also proven in humans (DOI 10.1038/bjc.2016.329).
Response 4.
We are aware that this association is not limited to dogs, we have simply used this as an example to outline the breath of conditions, and species, where sRNA dysfunction is associated with disease. The addition of a human reference is not required considering the context in which it is discussed.
Comment 5.
The statement in line 72 is not correct. MiRNAs can also activate gene expression – one example is DOI 10.1007/978-981-10-4310-9_6.
Response 5.
Our statement is correct. The reported instances of miRNA activation of gene expression is minute in comparison to the vast volume of on miRNA-directed repression of gene expression. This is a general review and therefore the discussion of such specific details is outside the scope of the review. In addition, no evidence of miRNA activation of gene expression exists in the algae on which this review article is primarily focused. Therefore, the inclusion of such a statement is not relevant to this review article.
Comment 6.
The authors have to explain why they are focusing on three model species (lines148-150).
Response 6.
We clearly outlined in the original manuscript why only three species are discussed. The miRNA pathway is exceptionally well charcaterised in plants and animals, and is starting to become reasonably well charcaterised in microalgae. We have therefore used a single species (where the majority of discoveries have been made) to discuss the miRNA pathways of the plant, animal and microalga systems.
Comment 7.
All abbreviations used in the figures must be explained below in the relevant figure legends.
Response 7.
We thank Reviewer 1 for this excellent suggestion, and we apologise for this oversight. An abbreviation list has been added to the legends of Figures 1, 2, 3 and 4.
Comment 8.
It is neither nice nor necessary to read “…. Molnar et al. .…”, “…Zhao et al.” etc. If somebody is interested in the name of the first author (s)he will find this information in the Reference list. The authors have to rephrase all these parts.
Response 8.
We have addressed this issue in the revised manuscript. We thank the Reviewer for this suggestion.
Comment 9.
Size of Figure 5 must be increased.
Response 9.
We thank Reviewer 1 for identifying this issue. We have reworked Figure 5 in the revised manuscript so that the Figure can be of an appropriate size. The authors thank Reviewer 1 for all of their insight and helpful suggestions.
Reviewer 2 Report
Comments and Suggestions for Authors
In the manuscript entitled “The microRNA pathway of macroalgae: its similarities and differences to the plant and animal microRNA pathways”, the authors described a comprehensive review of miRNAs in different organisms by focusing on macroalgae. The topic of the manuscript is interesting and it can gain the attention of researchers who work in this field. I believe this manuscript should consider the following major comments;
-Add some data about the presence of ncRNAs in micro and macroalgae and their biological functions.
-Add a few sentences about the possible interactions between miRNAs and ncRNAs since they impose their functions by acting as a sponge for ncRNAs like lncRNAs or circRNAs.
-Add some data for the presence and function of miRNAs in microalgae such as Dunaliella and Haematococcus.
-The English style of the manuscript should be improved.
Comments on the Quality of English LanguageThe English style of the manuscript should be improved.
Author Response
Responses to Reviewer 2 Comments.
Comment 1.
Add some data about the presence of ncRNAs in micro and macroalgae and their biological functions.
Response 1.
Our review is exclusively focused on the miRNA pathway, and to ensure that this focus is maintained throughout the review, we have almost solely centred our discussions around this class of small non-coding RNA. Supplying such information, considering its scarcity currently in the algae, and which does not form a focus of the review would only serve to dilute the primary focus of our review, and we do not wish to do this. We have however added a statement in the Acknowledgements section of the review detailing that we could not cover all aspects of small RNA biology in the review.
Comment 2.
Add a few sentences about the possible interactions between miRNAs and ncRNAs since they impose their functions by acting as a sponge for ncRNAs like lncRNAs or circRNAs.
Response 2.
This relates directly to Response 1 above. Next to no information currently exists for these types of RNA species in the seaweeds, nor the microalgae. The inclusion of any such text would therefore only serve to subtract from the main focus of our review article.
Comment 3.
Add some data for the presence and function of miRNAs in microalgae such as Dunaliella and Haematococcus.
Response 3.
We have added this data as requested by Reviewer 2, and we thank the reviewer for this helpful suggestion. Please see text lines 569 to 605 of the revised manuscript (pages 13 and 14).
Comment 4.
The English style of the manuscript should be improved.
Response 4.
We have performed an extensive edit to our manuscript’s text as part of preparing the revised manuscript. Reviewer 1 comments on the high standard of language used in our manuscript, and we are therefore of the opinion that the style of English language used in the authoring of the original manuscript was appropriate. Nonetheless, we have edited the text of the revised manuscript to ensure that a high standard is maintained between the two manuscript versions.